# Amortized Bethe Free Energy Minimization for Learning MRFs

**Sam Wiseman**
Toyota Technological Institute at Chicago
Chicago, IL, USA
swiseman@ttic.edu

**Yoon Kim**
Harvard University
Cambridge, MA, USA
yoonkim@seas.harvard.edu

## Abstract

We propose to learn deep undirected graphical models (i.e., MRFs) with a non-ELBO objective for which we can calculate exact gradients. In particular, we optimize a saddle-point objective deriving from the Bethe free energy approximation to the partition function. Unlike much recent work in approximate inference, the derived objective requires no sampling, and can be efficiently computed even for very expressive MRFs. We furthermore amortize this optimization with trained inference networks. Experimentally, we find that the proposed approach compares favorably with loopy belief propagation, but is faster, and it allows for attaining better held out log likelihood than other recent approximate inference schemes.

## 1 Introduction

There has been much recent work on learning deep generative models of discrete data, in both the case where all the modeled variables are observed [35, 58, *inter alia*], and in the case where they are not [37, 36, *inter alia*]. Most of this recent work has focused on directed graphical models, and when approximate inference is necessary, on variational inference. Here we consider instead undirected models, that is, Markov Random Fields (MRFs), which we take to be interesting for at least two reasons: first, some data are more naturally modeled using MRFs [25]; second, unlike their directed counterparts, many intractable MRFs of interest admit a learning objective which both approximates the log marginal likelihood, and which can be computed exactly (i.e., without sampling). In particular, log marginal likelihood approximations that make use of the Bethe Free Energy (BFE) [4] can be computed in time that effectively scales linearly with the number of factors in the MRF, provided that the factors are of low degree. Indeed, loopy belief propagation (LBP) [33], the classic approach to approximate inference in MRFs, can be viewed as minimizing the BFE [66]. However, while often quite effective, LBP is also an iterative message-passing algorithm, which is less amenable to GPU parallelization and can therefore slow down the training of deep generative models.

To address these shortcomings of LBP in the context of training deep models, we propose to train MRFs by minimizing the BFE directly during learning, without message-passing, using inference networks trained to output approximate minimizers. This scheme gives rise to a saddle-point learning problem, and we show that learning in this way allows for quickly training MRFs that are competitive with or outperform those trained with LBP.

We also consider the setting where the discrete latent variable model to be learned admits both directed and undirected variants. For example, we might be interested in learning an HMM-like model, but we are free to parameterize transition factors in a variety of ways, including such that all the transition factors are unnormalized and of low-degree (see Figure 1). Such a parameterization makes BFE minimization particularly convenient, and indeed we show that learning such an undirected model with BFE minimization allows for outperforming the directed variant learned with amortized variational inference in terms of both held out log likelihood and speed. Thus, when possible, it may

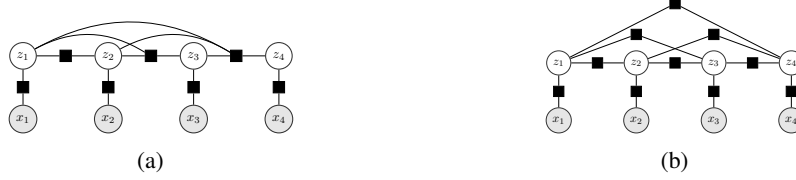

<center>(a)                      (b)</center>

Figure 1: Factor graphs of (a) a full 3rd order HMM, and (b) a 3rd order HMM-like model with only pairwise factors.

in fact be advantageous to consider transforming a directed model into an undirected variant, and learning it with BFE minimization.

## 2 Background

Let $\mathcal{G} = (\mathcal{V} \cup \mathcal{F}, \mathcal{E})$ be a factor graph [11, 26], with $\mathcal{V}$ the set of variable nodes, $\mathcal{F}$ the set of factor nodes, and $\mathcal{E}$ the set of undirected edges between elements of $\mathcal{V}$ and elements of $\mathcal{F}$; see Figure 1 for examples. We will refer collectively to variables in $\mathcal{V}$ that are always observed as $\mathbf{x}$, and to variables which are never observed as $\mathbf{z}$. We will take all variables to be discrete.

In a Markov Random Field (MRF), the joint distribution over $\mathbf{x}$ and $\mathbf{z}$ factorizes as $P(\mathbf{x}, \mathbf{z}; \boldsymbol{\theta}) = \frac{1}{Z(\boldsymbol{\theta})} \prod_{\alpha} \Psi_{\alpha}(\mathbf{x}_{\alpha}, \mathbf{z}_{\alpha}; \boldsymbol{\theta})$, where the notation $\mathbf{x}_{\alpha}$ and $\mathbf{z}_{\alpha}$ is used to denote the (possibly empty) subvectors of $\mathbf{x}$ and $\mathbf{z}$ that participate in factor $\Psi_{\alpha}$, the factors $\Psi_{\alpha}$ are assumed to be positive and are parameterized by $\boldsymbol{\theta}$, and where $Z(\boldsymbol{\theta})$ is the partition function: $Z(\boldsymbol{\theta}) = \sum_{\mathbf{x}'} \sum_{\mathbf{z}'} \prod_{\alpha} \Psi_{\alpha}(\mathbf{x}'_{\alpha}, \mathbf{z}'_{\alpha}; \boldsymbol{\theta})$.

In order to simplify the exposition we will assume all factors are either unary (functions of a single variable in $\mathcal{V}$) or pairwise (functions of two variables in $\mathcal{V}$), and we lose no generality in doing so [67, 60]. Thus, if a node $v_1 \in \mathcal{V}$ may take on one of $K_1$ discrete values, we view a unary factor $\Psi_{\alpha}(\mathbf{x}_{\alpha}, \mathbf{z}_{\alpha}; \boldsymbol{\theta}) = \Psi_{\alpha}(v_1; \boldsymbol{\theta})$ as a function $\Psi_{\alpha} : \{1, \ldots, K_1\} \to \mathbb{R}_+$. Similarly, if nodes $v_1$ and $v_2$ may take on $K_1$ and $K_2$ discrete values respectively, we view a binary factor $\Psi_{\beta}(\mathbf{x}_{\beta}, \mathbf{z}_{\beta}; \boldsymbol{\theta}) = \Psi_{\beta}(v_1, v_2; \boldsymbol{\theta})$ as a function $\Psi_{\beta} : \{1, \ldots, K_1\} \times \{1, \ldots, K_2\} \to \mathbb{R}_+$. It will also be convenient to use the (bolded) notation $\boldsymbol{\Psi}_{\alpha}$ to refer to the vector of a factor's possible output values (in $\mathbb{R}_+^{K_1}$ and $\mathbb{R}_+^{K_1 \cdot K_2}$ for unary and binary factors, respectively), and the notation $|\boldsymbol{\Psi}_{\alpha}|$ to refer to the length of this vector. We will consider both scalar and neural parameterizations of factors.

When the model involves unobserved variables, we will also make use of the "clamped" partition function $Z(\mathbf{x}, \boldsymbol{\theta}) = \sum_{\mathbf{z}'} \prod_{\alpha} \Psi_{\alpha}(\mathbf{x}_{\alpha}, \mathbf{z}'_{\alpha}; \boldsymbol{\theta})$, with $\mathbf{x}$ clamped to a particular value. The clamped partition function gives the unnormalized marginal probability of $\mathbf{x}$, the partition function of $P(\mathbf{z} \,|\, \mathbf{x}; \boldsymbol{\theta})$.

### 2.1 The Bethe Free Energy

Because calculation of $Z(\boldsymbol{\theta})$ or $Z(\mathbf{x}, \boldsymbol{\theta})$ may be intractable, maximum likelihood learning of MRFs often makes use of approximations to these quantities. One such approximation makes use of the Bethe free energy (BFE), due to Bethe [4] and popularized by Yedidia et al. [66], which is defined in terms of the factor and node marginals of the corresponding factor graph. In particular, let $\tau_{\alpha}(\mathbf{x}'_{\alpha}, \mathbf{z}'_{\alpha}) \in [0, 1]$ be the marginal probability of the event $\mathbf{x}'_{\alpha}$ and $\mathbf{z}'_{\alpha}$, which are again (possibly empty) settings of the subvectors associated with factor $\Psi_{\alpha}$. We will refer to the vector consisting of the concatenation of *all* possible marginals for each factor in $\mathcal{G}$ as $\boldsymbol{\tau} \in [0, 1]^{M(\mathcal{G})}$, where $M(\mathcal{G}) = \sum_{\alpha \in \mathcal{F}} |\boldsymbol{\Psi}_{\alpha}|$, the total number of values output by all factors associated with the graph. As a concrete example, consider the 10 factors in Figure 1 (b): if each variable can take on only two possible values, then since each factor is pairwise (i.e., considers only two variables), there are $2^2$ possible settings for each factor, and thus $2^2$ corresponding marginals. In total, we then have $10 \times 4$ marginals and so $\boldsymbol{\tau} \in [0, 1]^{40}$.

Following Yedidia et al. [67], the BFE is then defined as

$$F(\boldsymbol{\tau}, \boldsymbol{\theta}) = \sum_{\alpha} \sum_{\mathbf{x}'_{\alpha}, \mathbf{z}'_{\alpha}} \tau_{\alpha}(\mathbf{x}'_{\alpha}, \mathbf{z}'_{\alpha}) \log \frac{\tau_{\alpha}(\mathbf{x}'_{\alpha}, \mathbf{z}'_{\alpha})}{\Psi_{\alpha}(\mathbf{x}'_{\alpha}, \mathbf{z}'_{\alpha})} - \sum_{v \in \mathcal{V}} (|\mathrm{ne}(v)| - 1) \sum_{v'} \tau_v(v') \log \tau_v(v'), \quad (1)$$

<center>2</center>

where $\text{ne}(v)$ gives the set of factor-neighbors node $v$ has in the factor graph, and $\tau_v(v')$ is the marginal probability of node $v$ taking on the value $v'$.

Importantly, in the case of a distribution $P_{\boldsymbol{\theta}}$ representable as a tree-structured model, we have $\min_{\boldsymbol{\tau}} F(\boldsymbol{\tau}, \boldsymbol{\theta}) = -\log Z(\boldsymbol{\theta})$, since (1) is precisely $\text{KL}[Q||P_{\boldsymbol{\theta}}] - \log Z(\boldsymbol{\theta})$, where $Q$ is another tree representable distribution with marginals $\boldsymbol{\tau}$ [17, 60, 13]. In the case where $P_{\boldsymbol{\theta}}$ is *not* tree-structured (i.e., it has a loopy factor graph), we no longer have a KL divergence, and $\min_{\boldsymbol{\tau}} F(\boldsymbol{\tau}, \boldsymbol{\theta})$ will in general give only an approximation, but not a bound, on the partition function: $\min_{\boldsymbol{\tau}} F(\boldsymbol{\tau}, \boldsymbol{\theta}) \approx -\log Z(\boldsymbol{\theta})$ [60, 65, 61, 62].

Although minimizing the BFE only provides an approximation to $-\log Z(\boldsymbol{\theta})$, it is attractive for our purposes because while the BFE is exponential in the *degree* of each factor (since it sums over all assignments), it is only linear in the number of factors. Thus, evaluating (1) for a factor graph with a large number of small-degree (e.g., pairwise) factors remains tractable. Moreover, while restricting models to have low-degree factors severely limits the expressiveness of *directed* graphical models, it does not so limit the expressiveness of MRFs, since MRFs are free to have arbitrary pairwise dependence, as in Figure 1 (b). Indeed, the idea of establishing complex dependencies through many pairwise factors in an MRF is what underlies product-of-experts style modeling [18].

## 2.2 Minimizing the Bethe Free Energy

Historically, the BFE has been minimized during learning with loopy belief propagation (LBP) [41, 33]. Yedidia et al. [66] show that the fixed points found by LBP correspond to stationary points of the optimization problem $\min_{\boldsymbol{\tau} \in \mathcal{C}} F(\boldsymbol{\tau}, \boldsymbol{\theta})$, where $\mathcal{C}$ contains vectors of length $M(\mathcal{G})$, and in particular the concatenation of "pseudo-marginal" vectors $\boldsymbol{\tau}_\alpha(\mathbf{x}_\alpha, \mathbf{z}_\alpha)$ for each factor, subject to each pseudo-marginal vector being positive and summing to 1, and the pseudo-marginal vectors being locally consistent. Local consistency requires that the pseudo-marginal vectors associated with any two factors $\alpha, \beta$ sharing a variable $v$ agree: $\sum_{\mathbf{x}'_\alpha, \mathbf{z}'_\alpha \setminus v} \boldsymbol{\tau}_\alpha(\mathbf{x}'_\alpha, \mathbf{z}'_\alpha) = \sum_{\mathbf{x}'_\beta, \mathbf{z}'_\beta \setminus v} \boldsymbol{\tau}_\beta(\mathbf{x}'_\beta, \mathbf{z}'_\beta)$; see also Heskes [17]. Note that even if $\boldsymbol{\tau}$ satisfies these conditions, for loopy models it may still not correspond to the marginals of any distribution [60].

While LBP is quite effective in practice [33, 38, 67, 34], it does not integrate well with the current GPU-intensive paradigm for training deep generative models, since it is a typically sequential message-passing algorithm (though see Gonzalez et al. [12]), which may require a variable number of iterations and a particular message-passing scheduling to converge [10, 13]. We therefore propose to drop the message-passing metaphor, and instead directly minimize the constrained BFE during learning using inference networks [51, 23, 22, 56], which are trained to output approximate minimizers. This style of training gives rise to a saddle-point objective for learning, detailed in the next section.

## 3 Learning with Amortized Bethe Free Energy Minimization

Consider learning an MRF consisting of only observed variables $\mathbf{x}$ via maximum likelihood, which requires minimizing $-\log P(\mathbf{x}; \boldsymbol{\theta}) = -\log \tilde{P}(\mathbf{x}; \boldsymbol{\theta}) + \log Z(\boldsymbol{\theta})$, where $\log \tilde{P}(\mathbf{x}; \boldsymbol{\theta}) = \sum_\alpha \log \Psi_\alpha(\mathbf{x}_\alpha; \boldsymbol{\theta})$. Using the Bethe approximation to $\log Z(\boldsymbol{\theta})$ from the previous section, we then arrive at the objective:

$$\ell_F(\boldsymbol{\theta}) = -\log \tilde{P}(\mathbf{x}; \boldsymbol{\theta}) - \min_{\boldsymbol{\tau} \in \mathcal{C}} F(\boldsymbol{\tau}) \approx -\log \tilde{P}(\mathbf{x}; \boldsymbol{\theta}) + \log Z(\boldsymbol{\theta}), \qquad (2)$$

and thus the saddle-point learning problem:

$$\min_{\boldsymbol{\theta}} \ell_F(\boldsymbol{\theta}) = \min_{\boldsymbol{\theta}} \left[ -\log \tilde{P}(\mathbf{x}; \boldsymbol{\theta}) - \min_{\boldsymbol{\tau} \in \mathcal{C}} F(\boldsymbol{\tau}, \boldsymbol{\theta}) \right] = \min_{\boldsymbol{\theta}} \max_{\boldsymbol{\tau} \in \mathcal{C}} \left[ -\log \tilde{P}(\mathbf{x}; \boldsymbol{\theta}) - F(\boldsymbol{\tau}, \boldsymbol{\theta}) \right]. \quad (3)$$

While $\ell_F$ is neither an upper nor lower bound on $-\log P(\mathbf{x}; \boldsymbol{\theta})$, it is an approximation, and indeed its gradients are precisely those that arise from approximating the true gradient of $-\log P(\mathbf{x}; \boldsymbol{\theta})$ by replacing the factor marginals in the gradient with pseudo-marginals; see Sutton et al. [53].

In the case where our MRF contains unobserved variables $\mathbf{z}$, we wish to learn by minimizing $-\log Z(\mathbf{x}, \boldsymbol{\theta}) + \log Z(\boldsymbol{\theta})$. Here we can additionally approximate the clamped partition function $-\log Z(\mathbf{x}, \boldsymbol{\theta})$ using the BFE. In particular, we have $\min_{\boldsymbol{\tau}_{\mathbf{x}} \in \mathcal{C}_{\mathbf{x}}} F(\boldsymbol{\tau}_{\mathbf{x}}, \boldsymbol{\theta}) \approx -\log Z(\mathbf{x}, \boldsymbol{\theta})$, where $\boldsymbol{\tau}_{\mathbf{x}}$ contains the marginals of the MRF with its observed variables clamped to $\mathbf{x}$ (which is equivalent to replacing these variables with unary factors, and so $\boldsymbol{\tau}_{\mathbf{x}}$ will in general be smaller than $\boldsymbol{\tau}$). We thus

arrive at the following saddle point learning problem for MRFs with latent variables:

$$\min_{\boldsymbol{\theta}} \ell_{F,\mathbf{z}}(\boldsymbol{\theta}) = \min_{\boldsymbol{\theta}} \left[ \min_{\boldsymbol{\tau}_{\mathbf{x}} \in \mathcal{C}_{\mathbf{x}}} F(\boldsymbol{\tau}_{\mathbf{x}}, \boldsymbol{\theta}) - \min_{\boldsymbol{\tau} \in \mathcal{C}} F(\boldsymbol{\tau}, \boldsymbol{\theta}) \right] = \min_{\boldsymbol{\theta}, \boldsymbol{\tau}_{\mathbf{x}}} \max_{\boldsymbol{\tau} \in \mathcal{C}} \left[ F(\boldsymbol{\tau}_{\mathbf{x}}, \boldsymbol{\theta}) - F(\boldsymbol{\tau}, \boldsymbol{\theta}) \right]. \quad (4)$$

## 3.1 Inference Networks

Optimizing $\ell_F$ and $\ell_{F,\mathbf{z}}$ requires tackling a constrained, saddle-point optimization problem. While we could in principle optimize over $\boldsymbol{\tau}$ or $\boldsymbol{\tau}_{\mathbf{x}}$ directly, we found this optimization to be difficult, and we instead follow recent work [51, 23, 22, 56] in replacing optimization over the variables of interest with optimization over the parameters $\boldsymbol{\phi}$ of an inference network $f(\cdot; \boldsymbol{\phi})$ outputting the variables of interest. Thus, an inference network consumes a graph $\mathcal{G}$ and predicts a pseudo-marginal vector; we provide additional details below.

We also note that because our inference networks consume graphs they are similar to graph neural networks [47, 29, 24, 68, *inter alia*]. However, because we are interested in being able to quickly learn MRFs, our inference networks do not do any iterative message-passing style updates; they simply consume either a symbolic representation of the graph or, in the "clamped" setting, a symbolic representation of the graph together with the observed variables. We provide further details of our inference network parameterizations in Section 4 and in the Supplementary Material.

**Handling Constraints on Predicted Marginals**   The predicted pseudo-marginals output by our inference network $f$ must respect the positivity, normalization, and local consistency constraints described in Section 2.2. Since the normalization and local consistency constraints are linear equality constraints, it is possible to optimize only in the subspace they define. However, such an approach requires the explicit calculation of a basis for the null space of the constraint matrix, which becomes unwieldy as the graph gets large. We accordingly adopt the much simpler and more scalable approach of handling the positivity and normalization constraints by optimizing over the "softmax basis" (i.e., over logits), and we handle the local consistency constraints by simply adding a term to our objective that penalizes this constraint violation [7, 40].

In particular, let $\mathbf{f}(\mathcal{G}, \alpha; \boldsymbol{\phi}) \in \mathbb{R}^{K_1 \cdot K_2}$ be the vector of scores given by inference network $f$ to all configurations of variables associated with factor $\alpha$. We define the predicted factor marginals to be

$$\boldsymbol{\tau}_\alpha(\mathbf{x}_\alpha, \mathbf{z}_\alpha; \boldsymbol{\phi}) = \mathrm{softmax}(\mathbf{f}(\mathcal{G}, \alpha; \boldsymbol{\phi})). \quad (5)$$

We obtain predicted node marginals for each node $v$ by averaging all the associated factor-level marginals:

$$\boldsymbol{\tau}_v(v; \boldsymbol{\phi}) = \frac{1}{|\mathrm{ne}(v)|} \sum_{\alpha \in \mathrm{ne}(v)} \sum_{\mathbf{x}'_\alpha, \mathbf{z}'_\alpha \setminus v} \boldsymbol{\tau}_\alpha(\mathbf{x}'_\alpha, \mathbf{z}'_\alpha; \boldsymbol{\phi}). \quad (6)$$

We obtain our final learning objective by adding a term penalizing the distance between the marginal associated with node $v$ according to a *particular* factor, and $\boldsymbol{\tau}_v(v; \boldsymbol{\phi})$. Thus, the optimization problem (3) becomes

$$\min_{\boldsymbol{\theta}} \max_{\boldsymbol{\phi}} \left[ -\log \tilde{P}(\mathbf{x}; \boldsymbol{\theta}) - F(\boldsymbol{\tau}(\boldsymbol{\phi}), \boldsymbol{\theta}) - \frac{\lambda}{|\mathcal{F}|} \sum_{v \in \mathcal{V}} \sum_{\alpha \in \mathrm{ne}(v)} d\Big( \boldsymbol{\tau}_v(v; \boldsymbol{\phi}), \sum_{\mathbf{x}'_\alpha, \mathbf{z}'_\alpha \setminus v} \boldsymbol{\tau}_\alpha(\mathbf{x}'_\alpha, \mathbf{z}'_\alpha; \boldsymbol{\phi}) \Big) \right], \quad (7)$$

where $d(\cdot, \cdot)$ is a non-negative distance or divergence calculated between the marginals (typically $L_2$ distance in experiments), $\lambda$ is a tuning parameter, and the notation $\boldsymbol{\tau}(\boldsymbol{\phi})$ refers to the entire vector of concatenated predicted marginals. We note that the number of penalty terms in (7) scales with $|\mathcal{F}|$, since we penalize agreement with node marginals; an alternative objective that penalizes agreement between factor marginals is possible, but would scale with $|\mathcal{F}|^2$.

Finally, we note that we can obtain an analogous objective for the latent variable saddle-point problem (4) by introducing an additional inference network $f_{\mathbf{x}}$ which additionally consumes $\mathbf{x}$, and adding an additional set of penalty terms.

## 3.2 Learning

We learn by alternating $I_1$ steps of gradient ascent on (7) with respect to $\boldsymbol{\phi}$ with one step of gradient descent on (7) with respect to $\boldsymbol{\theta}$. When the MRF contains latent variables, we take $I_2$ gradient

**Algorithm 1** Saddle-point MRF Learning

---

**for** $i = 1, \ldots, I_1$ **do**
    Obtain $\boldsymbol{\tau}(\boldsymbol{\phi})$ from $f(\cdot; \boldsymbol{\phi})$ using Equations (5) and (6)
    $\boldsymbol{\phi} \leftarrow \boldsymbol{\phi} + \nabla_{\boldsymbol{\phi}}[-F(\boldsymbol{\tau}(\boldsymbol{\phi}), \boldsymbol{\theta}) - \frac{\lambda}{|\mathcal{F}|} \sum_{v \in \mathcal{V}} \sum_{\alpha \in \mathrm{ne}(v)} d(\boldsymbol{\tau}_v(v; \boldsymbol{\phi}), \sum_{\mathbf{x}'_\alpha, \mathbf{z}'_\alpha \backslash v} \boldsymbol{\tau}_\alpha(\mathbf{x}'_\alpha, \mathbf{z}'_\alpha; \boldsymbol{\phi}))]$
**if** there are latents **then**
    **for** $i = 1, \ldots, I_2$ **do**
        Obtain $\boldsymbol{\tau}_{\mathbf{x}}(\boldsymbol{\phi}_{\mathbf{x}})$ from $f_{\mathbf{x}}(\cdot; \boldsymbol{\phi}_{\mathbf{x}})$ using Equations (5) and (6)
        $\boldsymbol{\phi}_{\mathbf{x}} \leftarrow \boldsymbol{\phi}_{\mathbf{x}} - \nabla_{\boldsymbol{\phi}_{\mathbf{x}}}[F(\boldsymbol{\tau}_{\mathbf{x}}(\boldsymbol{\phi}_{\mathbf{x}}), \boldsymbol{\theta}) + \frac{\lambda}{|\mathcal{F}|} \sum_{v \in \mathbf{z}} \sum_{\alpha \in \mathrm{ne}(v)} d(\boldsymbol{\tau}_{\mathbf{x}}(v; \boldsymbol{\phi}_{\mathbf{x}}), \sum_{\mathbf{x}'_\alpha, \mathbf{z}'_\alpha \backslash v} \boldsymbol{\tau}_{\mathbf{x}, \alpha}(\mathbf{x}'_\alpha, \mathbf{z}'_\alpha; \boldsymbol{\phi}_{\mathbf{x}}))]$
    $\boldsymbol{\theta} \leftarrow \boldsymbol{\theta} - \nabla_{\boldsymbol{\theta}}[F(\boldsymbol{\tau}_{\mathbf{x}}(\boldsymbol{\phi}_{\mathbf{x}}), \boldsymbol{\theta}) - F(\boldsymbol{\tau}(\boldsymbol{\phi}), \boldsymbol{\theta})]$
**else**
    $\boldsymbol{\theta} \leftarrow \boldsymbol{\theta} - \nabla_{\boldsymbol{\theta}}[-\log \tilde{P}(\mathbf{x}; \boldsymbol{\theta}) - F(\boldsymbol{\tau}(\boldsymbol{\phi}), \boldsymbol{\theta})]$

---

descent steps to minimize the objective with respect to $\boldsymbol{\phi}_{\mathbf{x}}$ before updating $\boldsymbol{\theta}$. We show pseudo-code describing this procedure for a single minibatch in Algorithm 1.

Before moving on to experiments we emphasize two of the attractive features of the learning scheme described in (7) and Algorithm 1, which we verify empirically in the next section. First, because there is no message-passing and because minimization with respect to the $\boldsymbol{\tau}$ and $\boldsymbol{\tau}_{\mathbf{x}}$ pseudo-marginals is amortized using inference networks, we are often able to reap the benefits of training MRFs with LBP but much more quickly. Second, we emphasize that the objective (7) and its gradients can be calculated exactly, which stands in contrast to much recent work in variational inference for both directed models [43, 23] and undirected models [27], where the ELBO and its gradients must be approximated with sampling. As the variance of ELBO gradient estimators is known to be an issue when learning models with discrete latent variables [37], if it is possible to develop undirected analogs of the models of interest it may be beneficial to do so, and then learn these models with the $\ell_F$ or $\ell_{F,\mathbf{z}}$ objectives, rather than approximating the ELBO. We consider one such case in the next section.

## 4 Experiments

Our experiments are designed to verify that amortizing BFE minimization is an effective way of performing inference, that it allows for learning models that generalize, and that we can do this quickly. We accordingly consider learning and performing inference on three different kinds of popular MRFs, comparing amortized BFE minimization with standard baselines. We provide additional experimental details in the Supplementary Material, and code for duplicating experiments is available at `https://github.com/swiseman/bethe-min`.

### 4.1 Ising Models

We first study our approach as applied to Ising models. An $n \times n$ grid Ising model gives rise to a distribution over binary vectors $\mathbf{x} \in \{-1, 1\}^{n^2}$ via the following parameterization: $P(\mathbf{x}; \boldsymbol{\theta}) = \frac{1}{Z(\boldsymbol{\theta})} \exp(\sum_{(i,j) \in \mathcal{E}} J_{ij} x_i x_j + \sum_{i \in \mathcal{V}} h_i x_i)$, where $J_{ij}$ are the pairwise log potentials and $h_i$ are the node log potentials. The generative model parameters are thus given by $\boldsymbol{\theta} = \{J_{ij}\}_{(i,j) \in \mathcal{E}} \cup \{h_i\}_{i \in \mathcal{V}}$. While Ising models are conceptually simple, they are in fact quite general since any binary pairwise MRF can be transformed into an equivalent Ising model [50].

In these experiments, we are interested in quantifying how well we can approximate the true marginal distributions with approximate marginal distributions obtained from the inference network. We therefore experiment with model sizes for which exact inference is reasonably fast on modern hardware (up to $15 \times 15$).[1]

Our inference network associates a learnable embedding vector $\mathbf{e}_i$ with each node and applies a single Transformer layer [59] to obtain a new node representation $\mathbf{h}_i$, with $[\mathbf{h}_1, \ldots, \mathbf{h}_{n^2}] = \mathrm{Transformer}([\mathbf{e}_1, \ldots, \mathbf{e}_{n^2}])$. The distribution over $x_i, x_j$ for $(i,j) \in \mathcal{E}$ is given by concatenating $\mathbf{h}_i, \mathbf{h}_j$ and applying an affine layer followed by a softmax: $\boldsymbol{\tau}_{ij}(x_i, x_j; \boldsymbol{\phi}) = \mathrm{softmax}(\mathbf{W}[\mathbf{h}_i; \mathbf{h}_j] + \mathbf{b})$. The parameters of the inference network $\boldsymbol{\phi}$ are given by the node embed-

Table 1: Correlation and Mean $L_1$ distance between the true vs. approximated marginals for the various methods.

| | Correlation | | | Mean $L_1$ distance | | |
|---|---|---|---|---|---|---|
| $n$ | Mean Field | Loopy BP | Inference Network | Mean Field | Loopy BP | Inference Network |
| 5 | 0.835 | 0.950 | 0.988 | 0.128 | 0.057 | 0.032 |
| 10 | 0.854 | 0.946 | 0.984 | 0.123 | 0.064 | 0.037 |
| 15 | 0.833 | 0.942 | 0.981 | 0.132 | 0.065 | 0.040 |

Figure 2: For each method, we plot the approximate marginals (x-axis) against the true marginals (y-axis) for a $15 \times 15$ Ising model. Top shows the node marginals while bottom shows the pairwise factor marginals, and $\rho$ denotes the Pearson correlation coefficient.

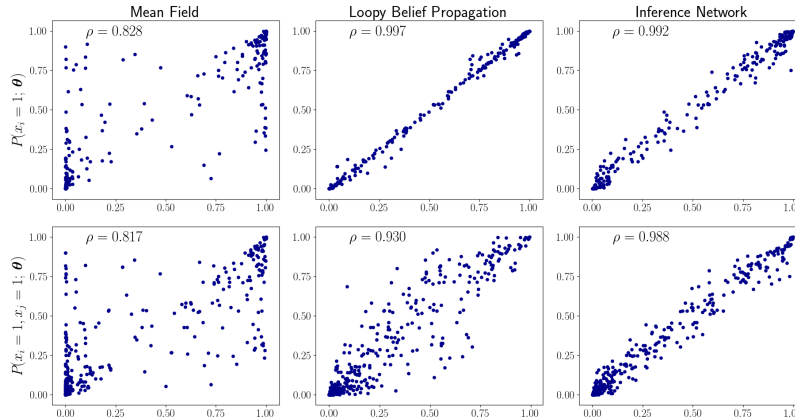

dings and the parameters of the Transformer/affine layers. The node marginals $\boldsymbol{\tau}_i(x_i; \boldsymbol{\phi})$ then are obtained from averaging the pairwise factor marginals (Eq (6)).[2]

We first examine whether minimizing the BFE with an inference network gives rise to reasonable marginal distributions. Concretely, for a fixed $\boldsymbol{\theta}$ (sampled from spherical Gaussian with unit variance), we minimize $F(\boldsymbol{\tau}(\boldsymbol{\phi}), \boldsymbol{\theta})$ (Eq (1)) with respect to $\boldsymbol{\phi}$, where $\boldsymbol{\tau}(\boldsymbol{\phi})$ denotes the full vector of marginal distributions obtained from the inference network. Table 1 shows the correlation and the mean $L_1$ distance between the true marginals and the approximated marginals, where the numbers are averaged over 100 samples of $\boldsymbol{\theta}$. We find that compared to approximate marginals obtained from mean field and LBP the inference network produces marginal distributions that are more accurate. Figure 2 shows a scatter plot of approximate marginals ($x$-axis) against the true marginals ($y$-axis) for a randomly sampled $15 \times 15$ Ising model. Interestingly, we observe that both loopy belief propagation and the inference network produce accurate node marginals (top), but the pairwise factor marginals from the inference network are much better (bottom). We find that this trend holds for Ising models with greater pairwise interaction strength as well; see the additional experiments in the Supplementary Material where pairwise potentials are sampled from $\mathcal{N}(0, 3)$ and $\mathcal{N}(0, 5)$.

In Table 2 we show results from learning the generative model alongside the inference network. For a randomly generated Ising model, we obtain 1000 samples each for train, validation, and test sets, using a version of the forward-filtering backward-sampling algorithm to obtain exact samples in $O(2^n)$. We then train a (randomly-initialized) Ising model via the saddle point learning problem in Eq (7). While models trained with exact inference perform best, models trained with an inference network's approximation to $\log Z(\boldsymbol{\theta})$ perform almost as well, and outperform both those trained with mean field and even with LBP. See the Supplementary Material for additional details.

## 4.2 Restricted Boltzmann Machines (RBMs)

We next consider learning Restricted Boltzmann Machines [49], a classic MRF model with latent variables. A binary RBM parameterizes the joint distribution over observed variables $\mathbf{x} \in \{0, 1\}^V$

Table 2: Held out NLL of learned Ising models. True entropy refers to NLL under the true model (i.e. $\mathbb{E}_{P(\mathbf{x};\boldsymbol{\theta})}[-\log P(\mathbf{x};\boldsymbol{\theta})])$, and 'Exact' refers to an Ising model trained with the exact partition function.

| $n$ | True Entropy | Rand. Init. | Exact | Mean Field | Loopy BP | Inference Network |
|-----|--------------|-------------|-------|------------|----------|-------------------|
| 5   | 6.27         | 45.62       | 6.30  | 7.35       | 7.17     | 6.47              |
| 10  | 25.76        | 162.53      | 25.89 | 29.70      | 28.34    | 26.80             |
| 15  | 51.80        | 365.36      | 52.24 | 60.03      | 59.79    | 54.91             |

Table 3: Held out average NLL of learned RBMs, as estimated by AIS [46]. Neural Variational Inference results are taken from Kuleshov and Ermon [27].

|                              | NLL      | $\ell_F$ | Epochs to Converge | Seconds/Epoch |
|------------------------------|----------|----------|--------------------|---------------|
| Loopy BP                     | 25.47    | 53.02    | 8                  | 21617         |
| Inference Network            | 23.43    | 23.11    | 38                 | 14            |
| PCD                          | 21.24    | N/A      | 29                 | 1             |
| Neural Variational Inference [27] | $\geq 24.5$ |          |                    |               |

and latent variables $\mathbf{z} \in \{0,1\}^H$ as $P(\mathbf{x}, \mathbf{z}; \boldsymbol{\theta}) = \frac{1}{Z(\boldsymbol{\theta})} \exp(\mathbf{x}^\top \mathbf{W} \mathbf{z} + \mathbf{x}^\top \mathbf{b} + \mathbf{z}^\top \mathbf{a})$. Thus, there is a pairwise factor for each $(x_i, z_j)$ pair, and a unary factor for each $x_i$ and $z_j$.

It is standard when learning RBMs to marginalize out the latent variables, which can be done tractably due to the structure of the model, and so we may train with the objective in (7). Our inference network is similar to that used in our Ising model experiments: we associate a learnable embedding vector with each node in the model, which we concatenate with an embedding corresponding to an indicator feature for whether the node is in $\mathbf{x}$ or $\mathbf{z}$. These $V + H$ embeddings are then consumed by a bidirectional LSTM [20, 15], which outputs vectors $\mathbf{h}_{\mathbf{x},i}$ and $\mathbf{h}_{\mathbf{z},j}$ for each node.[3] Finally, we obtain $\tau_{ij}(x_i, z_j; \boldsymbol{\phi}) = \mathrm{softmax}(\mathrm{MLP}[\mathbf{h}_{\mathbf{x},i}; \mathbf{h}_{\mathbf{z},j}])$. We set the $d(\cdot, \cdot)$ penalty function to be the KL divergence, which worked slightly better than L2 distance in preliminary experiments.

We follow the experimental setting of Kuleshov and Ermon [27], who recently introduced a neural variational approach to learning MRFs, and train RBMs with 100 hidden units on the UCI digits dataset [1], which consists of $8 \times 8$ images of digits. We compare with persistent contrastive divergence (PCD) [54] and LBP, as well as with the best results reported in Kuleshov and Ermon [27].[4] We used a batch size of 32, and selected hyperparameters through random search, monitoring validation expected pseudo-likelihood [3] for all models; see the Supplementary Material.

Table 3 reports the held out average NLL as estimated with annealed importance sampling (AIS) [39, 46], using 10 chains and $10^3$ intermediate distributions; it also reports average seconds per epoch, rounded to the nearest second.[5] We see that while amortized BFE minimization is able to outperform all results except PCD, it does lag behind PCD. These results are consistent with previous claims in the literature [46] that LBP and its variants do not work well on RBMs. Amortizing BFE minimization does, however, again outperform LBP. We also emphasize that PCD relies on being able to do fast block Gibbs updates during learning, which will not be available in general, whereas amortized BFE minimization has no such requirement.

## 4.3 High-order HMMs

Finally, we consider a scenario where both $Z(\boldsymbol{\theta})$ and $Z(\mathbf{x}, \boldsymbol{\theta})$ must be approximated, namely, that of learning 3rd order neural HMMs [55] (as in Figure 1) with *approximate* inference. We consider this setting in particular because it allows for the use of dynamic programs to compare the true NLL attained when learning with approximate inference. However, because these dynamic programs scale as $O(TK^{L+1})$, where $T, L, K$ are the sequence length, Markov order, and number of latent state values, respectively, considering even higher-order models becomes difficult. A standard 3rd order neural HMM parameterizes the joint distribution over observed sequence $\mathbf{x} \in \{1, \ldots, V\}^T$ and latent sequence $\mathbf{z} \in \{1, \ldots, K\}^T$ as $P(\mathbf{x}, \mathbf{z}; \boldsymbol{\theta}) = \frac{1}{Z(\boldsymbol{\theta})} \exp(\sum_{t=1}^T \log \Psi_{t,1}(z_{t-3:t}; \boldsymbol{\theta}) + \log \Psi_{t,2}(z_t, x_t; \boldsymbol{\theta}))$.

**Directed 3rd Order HMMs** To further motivate the results of this section let us begin by considering using approximate inference techniques to learn *directed* 3rd order neural HMMs, which are obtained by having each factor output a normalized distribution. In particular, we define the emission distribution $\boldsymbol{\Psi}_{t,2}(z_t{=}k, x_t; \boldsymbol{\theta}) = \mathrm{softmax}(\mathbf{W}\,\mathrm{LayerNorm}(\mathbf{e}_k + \mathrm{MLP}(\mathbf{e}_k)))$, where $\mathbf{e}_k \in \mathbb{R}^d$ is an embedding corresponding to the $k$'th discrete value $z_t$ can take on, $\mathbf{W} \in \mathbb{R}^{V \times d}$ is a word embedding matrix with a row for each word in the vocabulary, and layer normalization [2] is used to stabilize training. We also define the transition distribution $\boldsymbol{\Psi}_{t,1}(z_t, z_{t-1}{=}k_1, z_{t-2}{=}k_2) = \mathrm{softmax}(\mathbf{U}\,\mathrm{LayerNorm}([\mathbf{e}_{k_1}; \mathbf{e}_{k_2}]+\mathrm{MLP}([\mathbf{e}_{k_1}; \mathbf{e}_{k_2}])))$, where $\mathbf{U} \in \mathbb{R}^{K \times 2K}$ and the $\mathbf{e}_k$ are shared with the emission parameterization.

We now consider learning a $K = 30$ state 3rd order directed neural HMM on sentences from the Penn Treebank [32] (using the standard splits and preprocessing by Mikolov et al. [35]) of length at most 30. The top part of Table 4 compares the average NLL on the validation set obtained by learning such an HMM with exact inference against learning it with several variants of discrete VAE [43, 23] and the REINFORCE [64] gradient estimator. In particular, we consider two inference network architectures:

- Mean Field: we obtain approximate posteriors $q(z_t \,|\, x_{1:T})$ for each timestep $t$ as $\mathrm{softmax}(\mathbf{Q}\,\mathrm{LayerNorm}(\mathbf{e}_{x_t} + \mathbf{h}_t))$, where $\mathbf{h}_t \in \mathbb{R}^{d_2}$ is the output of a bidirectional LSTM [19, 15] run over the observations $x_{1:T}$, $\mathbf{e}_{x_t}$ is the embedding of token $x_t$, and $\mathbf{Q} \in \mathbb{R}^{K \times d_2}$.

- 1st Order: Instead of assuming the approximate posterior $q(z_{1:T} \,|\, x_{1:T})$ factorizes independently over timesteps, we assume it is given by the posterior of a first-order (and thus more tractable) HMM. We parameterize this inference HMM identically to the neural directed HMM above, except that it conditions on the observed sequence $x_{1:T}$ by concatenating the averaged hidden states of a bidirectional LSTM run over the sequence onto the $\mathbf{e}_k$.

For the mean field architecture we consider optimizing either the ELBO with the REINFORCE gradient estimator together with an input dependent baseline [37] for variance reduction, or the corresponding 10-sample IWAE objective [5]. When the 1st Order HMM inference network is used, we sample from it exactly using quantities calculated with the forward algorithm [42, 6, 48, 69]. We provide more details in the Supplementary Material.

As the top of Table 4 shows, exact inference significantly outperforms the approximate methods, perhaps due to the difficulty in controlling the variance of the ELBO gradient estimators.

**Undirected 3rd Order HMMs** An alternative to learning a 3rd order HMM with variational inference, then, is to consider an analogous *undirected* model, which can be learned using BFE approximations, and therefore requires no sampling. In particular, we will consider the 3rd order undirected product-of-experts style HMM in Figure 1 (b), which contains only pairwise factors, and parameterizes the joint distribution of $\mathbf{x}$ and $\mathbf{z}$ as $P(\mathbf{x}, \mathbf{z}; \boldsymbol{\theta}) = \frac{1}{Z(\boldsymbol{\theta})} \exp(\sum_{t=1}^{T} \sum_{s=\max(t-3,1)}^{t-1} \log \Psi_{t,1,s}(z_s, z_t; \boldsymbol{\theta}) + \sum_{t=1}^{T} \log \Psi_{t,2}(z_t, x_t; \boldsymbol{\theta}))$. Note that while this variant captures only a subset of the distributions that can be represented by the full parameterization (Figure 1 (a)), it still captures 3rd order dependencies using pairwise factors.

In our undirected parameterization the transition factors $\Psi_{t,1,s}$ are homogeneous (i.e., independent of the timestep) in order to allow for a fair comparison with the standard directed HMM, and are given by $\mathbf{r}_{k_2}^\top \mathrm{LayerNorm}([\mathbf{a}_{|t-s|}; \mathbf{e}_{k_1}] + \mathrm{MLP}([\mathbf{a}_{|t-s|}; \mathbf{e}_{k_1}]))$, where $\mathbf{a}_{|t-s|}$ is the embedding vector corresponding to factors relating two nodes that are $|t - s|$ steps apart, and where $\mathbf{e}_{k_1}$ and $\mathbf{r}_{k_2}$ are again discrete state embedding vectors. The emission factors $\Psi_{t,2}$ are those used in the directed case.

We train inference networks $f$ and $f_{\mathbf{x}}$ to output pseudo-marginals $\boldsymbol{\tau}$ and $\boldsymbol{\tau}_{\mathbf{x}}$ as in Algorithm 1, using $I_1 = 1$ and $I_2 = 1$ gradient updates per minibatch. Because $Z(\boldsymbol{\theta})$ and $Z(\mathbf{x}, \boldsymbol{\theta})$ depend only on the latent variables (since factors involving the $x_t$ remain locally normalized), $f$ and $f_{\mathbf{x}}$ are bidirectional LSTMs consuming embeddings corresponding to the $z_t$, where $f_{\mathbf{x}}$ also consumes $\mathbf{x}$. In particular, $f_{\mathbf{x}}$ is almost identical to the mean field inference network described above, except it additionally consumes an embedding for the current node (as did the RBM and Ising model inference networks) and an embedding indicating the total number of nodes in the graph. The inference network $f$ producing unclamped pseudo-marginals is identical, except it does not consume $\mathbf{x}$. As the bottom of

Table 4: Average NLL of 3rd Order HMM variants learned with approximate and exact inference.

|  |  | NLL | -ELBO/$\ell_{F,\mathbf{z}}$ | Epochs to Converge | Seconds/Epoch |
|---|---|---|---|---|---|
| Directed | Exact | 105.66 | 105.66 | 20 | 137 |
|  | Mean-Field VAE + BL | 119.27 | 175.46 | 14 | 82 |
|  | Mean-Field IWAE-10 | 119.20 | 167.71 | 5 | 876 |
|  | 1st Order HMM VAE | 118.35 | 118.88 | 12 | 187 |
| Undirected | Exact | 104.07 | 104.07 | 20 | 122 |
|  | LBP | 108.74 | 99.89 | 20 | 247 |
|  | Inference Network | 115.86 | 114.75 | 11 | 70 |

Table 4 shows, this amortized approach manages to outperform all the VAE variants both in terms of held out average NLL and speed. It performs less well than true LBP, but is significantly faster.

## 5 Related Work

Using neural networks to perform approximate inference is a popular way to learn deep generative models, leading to a family of models called variational autoencoders [23, 44, 37]. However, such methods have generally been employed in the context of learning directed graphical models. Moreover, applying amortized inference to learn discrete latent variable models has proved challenging due to potentially high-variance gradient estimators that arise from sampling, though there have been some recent advances [21, 31, 57, 14].

Outside of directed models, several researchers have proposed to incorporate deep networks directly into message-passing inference operations, mostly in the context of computer vision applications. Heess et al. [16] and Lin et al. [30] train neural networks that learn to map input messages to output messages, while inference machines [45, 9] also directly estimate messages from inputs. In contrast, Li and Zemel [28] and Dai et al. [8] instead approximate iterations of mean field inference with neural networks.

Closely related to our work, Yoon et al. [68] employ a deep network over an underlying graphical model to obtain node-level marginal distributions. However, their inference network is trained against the true marginal distribution (i.e., not Bethe free energy as in the present work), and is therefore not applicable to settings where exact inference is intractable (e.g. RBMs). Also related is the early work of Welling and Teh [63], who also consider direct (but unamortized) minimization of the BFE, though only for inference and not learning. Finally, Kuleshov and Ermon [27] also learn undirected models via a variational objective, cast as an upper bound on the partition function.

## 6 Conclusion

We have presented an approach to learning MRFs which amortizes the minimization of the Bethe free energy by training inference networks to output approximate minimizers. This approach allows for learning models that are competitive with loopy belief propagation and other approximate inference schemes, and yet takes less time to train.

**Acknowledgments**

We are grateful to Alexander M. Rush and Justin Chiu for insightful conversations and suggestions. YK is supported by a Google AI PhD Fellowship.

## Footnotes

[1] The calculation of the partition function in grid Ising models is exponential in $n$, but it is possible to reduce the running time from $O(2^{n^2})$ to $O(2^n)$ with dynamic programming (i.e., variable elimination).

[2]As there are no latent variables in these experiments, inference via the inference network is not amortized in the traditional sense (i.e., across different data points as in Eq (4)) since it does not condition on $\mathbf{x}$. However, inference is still amortized across each optimization step, and thus we still consider this to be an instance of amortized inference.

[3]We found LSTMs to work somewhat better than Transformers for both the RBM and HMM experiments.

[4]The corresponding NLL number reported in Table 3 is derived from a figure in Kuleshov and Ermon [27].

[5]While it is difficult to exactly compare the speed of different learning algorithms, speed results were measured on the same 1080 Ti GPU, averaged over 10 epochs, and used our fastest implementations.

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
