[Supplementary Material]

# Supplementary Material for Amortized Bethe Free Energy Minimization for Learning MRFs

**Additional Details on Ising Model Experiments**

**Inference Network Details**    The $\mathbf{h}_i, \mathbf{h}_j$ calculated by the Transfomer layer were of size 200, as were the embeddings they consume. We note that if we view $\boldsymbol{\tau}_{ij}(x_i, x_j; \boldsymbol{\phi})$ as a $2 \times 2$ matrix over the four possible events, under our parameterization we do not necessarily have that $\boldsymbol{\tau}_{ij}(x_i, x_j; \boldsymbol{\phi}) = \boldsymbol{\tau}_{ij}(x_j, x_i; \boldsymbol{\phi})^\top$. We avoid this issue by calculating $\boldsymbol{\tau}_{ij}(x_i, x_j; \boldsymbol{\phi})$ only for $i < j$ (under a row-column ordering of the $n \times n$ grid), and then using $\boldsymbol{\tau}_{ij}(x_i, x_j; \boldsymbol{\phi})^\top$ for $j < i$.

**Additional Ising Model Experiments**

We additionally experiment with inference in Ising models with greater interaction strength, by sampling pairwise potentials from $\mathcal{N}(0, 3)$ and $\mathcal{N}(0, 5)$; unary potentials are still sampled from $\mathcal{N}(0, 1)$. We show the correlations in Table 1.

Table 1: Correlations between true and approximated marginals for Ising models with greater pairwise interaction strength.

| $n$ | $\sigma = 3$ | | | $\sigma = 5$ | | |
|---|---|---|---|---|---|---|
| | Mean Field | Loopy BP | Inf. Net | Mean Field | Loopy BP | Inf. Net |
| 5 | 0.415 | 0.592 | 0.761 | 0.302 | 0.509 | 0.614 |
| 10 | 0.460 | 0.641 | 0.770 | 0.358 | 0.525 | 0.619 |
| 15 | 0.435 | 0.665 | 0.738 | 0.362 | 0.459 | 0.564 |

**Training Details**    The inference network was trained with up to $I_1 = 200$ gradient steps to minimize the BFE with respect to $\boldsymbol{\tau}$, though optimization cut off early if the squared change in predicted pseudo-marginals was less than $10^{-5}$. Typically this tolerance was met after around 50 updates. Our LBP implementation was given the same budget.

**Additional Details on RBM Experiments**

**Inference Network Details**    We associated a 150-dimensional embedding with each node in the graph; we also embedded an indicator feature corresponding to whether a node is visible or hidden in 150 dimensional space. These embeddings were concatenated and fed into a 5-layer, 200-unit bidirectional LSTM, which consumed the embeddings first of the visible nodes, ordered row-wise, and then the hidden units. A two-layer MLP with ReLU nonlinearity was then used to predict the pseudo marginals for each edge, by consuming the corresponding top-level LSTM states. $\lambda$ was set to 1.5. We found that using a Transformer-based inference network performed slightly worse. Whereas all non-LBP methods were given a budget of 200 random hyperparameter configurations, LBP was tuned by hand due to its exorbitant runtime.

**Training Details**    We trained by doing only a single (i.e., $I_1 = 1$) update on the $\boldsymbol{\phi}$ parameters for every $\boldsymbol{\theta}$. Using more updates typically led to faster convergence but not improved results. LBP was allowed up to 10 full sweeps over all the nodes in the graph per iteration; messages were ordered randomly. LBP was also cut off early if messages changed by less than $10^{-3}$ on average.

**Additional Details on HMM Experiments**

We again used a random search to choose the hyperparameters for each model and for each training regime that minimized held out NLL, as evaluated with a dynamic program. This search considered embeddings and hidden states of dimensionality $\{64, 100, 150, 200\}$, between 1 and 4 layers for the inference network, learning rates, $\lambda$ penalties, and the random seed.

For the directed models, we obtained our best results by setting the $\mathbf{e}_k$ state embeddings to be 200-dimensional for the models learned with exact inference and the first-order VAE, and to be 100-dimensional for models learned with the mean field VAEs. The best word embedding sizes were 64, 100, and 64 dimensional for the first-order, baseline, and IWAE VAEs, respectively; their BLSTMS were of sizes 3x100, 3x200, and 2x100, respectively.

For the undirected models, we obtained our best results by setting the $\mathbf{e}_k$ to be 200 for the model learned with exact inference, and 64 for the models learned with LBP and amortized BFE minimization. The best inference network used 150-dimensional word embeddings, a 1x100 BLSTM, and $\lambda = 1$.

As in the RBM setting, preliminary experiments suggested that setting the penalty function $d(\cdot, \cdot)$ to be the KL divergence slightly outperformed L2 distance, and that BLSTM inference networks slightly outperformed Transformers.

**Training Details**  We trained with a batch size of 32. We again found that while we could speed up convergence by increasing $I_1$ and $I_2$ it did not lead to better performance.

LBP was given up to 5 full sweeps over all the nodes in the graph per iteration, but was cut off early if messages changed by less than $10^{-3}$. Here, unsurprisingly, we found a left-to-right ordering of messages to outperform random ordering.

All the aforementioned experiments on Ising Models, RBMs, and HMMs used Adam [1] for optimization.

# References

[1] Diederik P Kingma and Jimmy Ba. Adam: A method for stochastic optimization. *arXiv preprint arXiv:1412.6980*, 2014.