[Reviews · NeurIPS 2019]

Reviewer 1



The paper is well structured and well written and by and large reproducible. The main contribution is the proposal to replace message passing based algorithms like belief propagation (which are operating on the unknown marginals as independent variables) by parametrised graph neural networks encoding the marginals. This is to my knowledge novel. The authors show in their experiments, that their approach has better estimation capabilities than belief propagation. On the other hand, it is known how to solve the Bethe free energy optimisation problem by double loop algorithms. I would have expected a comparison with such methods in terms of estimation accuracy. Moreover, the approximation itself is known to be inherently weak in estimating pairwise and higher order marginals of an MRF, because they are estimated by the corresponding edge (hyper-edge) factors of the model up to unary multiplicands. This ignores the correlation of variables induced via other paths connecting the nodes in the graph. Even after reading the supplementary material, I am missing details about the inference networks an especially an explanation of what is meant by "the are run once ... over the symbolic representation of the graph ..." The experiments have in my view a preliminary character and, some details of them are missing: - Ising model: The interactions strength used for the experiments with the Ising MRFs remains unclear (pairwise potentials "sampled from a spherical Gaussian"). It is to expect that the estimation quality will degrade with the (average) interaction strength. Also, comparing the method to mean field approximation is meaningless for pairwise marginals(?) because the latter estimates unary marginals only. As already mentioned above, I would have expected comparison with known double loop algorithms for solving the Bethe free energy optimisation task. It would be valuable to show that the proposed method, being faster then double loop algorithms, gives similar approximation quality. - Higher order HMMs: The details given in the supplementary material are essential for understanding these experiments. The text given in the paper (paragraph starting from l. 253) is completely obscure. Summarising, while the proposal to amortize the marginals in terms of graph neural networks when solving the Bethe free energy approximation embedded in MLE learning of MRFs, is indeed interesting. The validation of this method, however, remains in my view preliminary only. Post-rebuttal comments: The authors addressed and answered almost all issues raised in my review. I will raise my positive mark.

Reviewer 2



- Overall the paper provides a creative new approach to learning/inference for Markov random fields and provides a thorough empirical study of its behavior. The writing is very clear and notation is precise. - The approach draws inspiration from a variety of existing work and ties together ideas from both classical graphical models work and modern deep MRF learning. The key idea is to use an inference network that approximates the factor/variable marginals of the MRF and to obtain its parameters by minimizing the (approximate) BFE by gradient descent. This inference technique can be incorporated into learning via a saddle point min-max problem. The gradient based learning algorithm alternates between updating the parameters of the MRF are to minimize the negative log-likelihood and updating the parameters of the inference network to maximize the negative BFE. The simplicity of this approach and the fact that the inference network can be some GPU-friendly deep net makes this potentially faster than LBP. Although the resulting algorithm provides no bounds on the partition function (akin to LBP), its results suggest that it is a promising solution. The paper plays a nice trick by enforcing a local consistency constraint via a penalty term on the node marginals in order to maintain an algorithm that is linear (instead of quadratic) in the number of factors. - The experiments suggest that the approach is more efficient (computationally) than LBP and that it often learns to predict more accurate marginals than learning with other approximate inference algorithms (e.g. LBP, mean field). The paper explores approximations of both the fully-observed and partially-observed (i.e. marginal) likelihood. The Ising model experiments are carried out on synthetic data on models for which comparisons to exact inference (by variable elimination) are tractable. The RBM experiments compare on the UCI digits dataset with recent work on Neural Variational Inference (Kuleshov and Ermon, 2017). The HMM experiments explore both directed and underdirected equivalents of HMMs that include neural factors (a la. Tran et al.) and evaluate on a odd (i.e. 30 tags instead of 45, short sentences), but previously used POS tagging setting on the Penn Treebank -- this permits comparison with other VAE approaches on directed HMMs. - While the pairwise MRF doesn't reduce the generality of this approach, does doing so have potentially negative impacts on runtime? That is, might one prefer working with some (slightly) higher-order factors for some efficiency gains in the inference network? - It would be great if the statement on line 134-136 were expanded upon in the supplementary material. What would it look like to optimize in that subspace? Are there cases where it might actually be a good idea (even if more complicated to do so)? - Empirically, how well does the penalty term succeed in enforcing local consistency? - How important is using gradient descent (as opposed to say batch SGD)? That is, does learning become unstable if you are not amortizing over the training set? - Missed citation: In line 166, the paper notes "Second, we emphasize that...its gradients can be calculated exactly, which stands in contrast to much recent work". Stoyanov et al (AISTATS 2011) "Empirical Risk Minimization of Graphical Model Parameters Given Approximate Inference, Decoding, and Model Structure" presents an alternative LBP-based solution that also computes exact gradients of an approximate system. - Why are the inference networks different between the Ising and RBM experiments? Did the transformer or biLSTM perform better/worse on or the other? - Somewhat disappointingly the paper never mentions what hardware was used.

Reviewer 3



The paper is well-written. The introduced approach is interesting, but the experimental results are not very impressive. 1) How amenable this traditionally difficult saddle point objective would be comparing to VI based approaches on larger datasets? 2) The marginal constraints can also be enforced by matrix operations. It would be interesting to contrast the two approaches. 3) The numbers in Table 1 and Figure 2 seem inconsistent. The figure suggests individual marginals of LBP are better. Is this due to the figure being a single sample or some other thing I am missing? 4) Given the sizes of the dataset considered I wonder whether some architecture of inference networks are better than others (the authors mentioned Transformers and RNN). It would be good to see the results across them. More questions about experiment details: 1) The form of discrepancy used for constraining the marginals is not clear. Did you use total variation distance, KL divergence or some other discrepancy measure? 2) In the appendix, the authors state that they have treated the random seed of the model also as a hyper-parameter. While it is good to see runs across different seeds, it is not clear to me why the seed has been treated as a hyperparameter. 3) I tried to run the provided code on ptb with the settings similar to that in the paper, and did not get numbers similar to the reported numbers. A glance at the code seems to suggest that the authors probably ran over 100k seeds. This coupled with the previous point makes me uncertain about the stability/reproducibility of the work. I would like the authors to provide the exact parameters with which they got these results. =========== I have read and considered the authors' response and the other reviews.

[Author Response · NeurIPS 2019]

We thank the reviewers for their insightful feedback; we address each review below.

**R1: "...it is known how to solve the BFE optimisation problem by double loop algorithms"** Our understanding is that double-loop algorithms (Yuille, 2001) converge to a local minimum, but not necessarily a global one. Further, double-loop algorithms outperform LBP generally when LBP does not converge, but in our experiments LBP converged reliably, and we thus anticipate that double-loop optimization will not necessarily improve further. We also note that despite the superiority of double loop algorithms, LBP remains popular, and so amortizing it seems useful in our view.

**"...what is meant by 'they are run once...'"** A Transformer or RNN is run over a sequence of embeddings corresponding to the nodes in the graph to obtain a sequence of representations. We do *not* then further update these Transformer/RNN representations with message-passing style updates as in much recent work in graph neural networks.

**"...meaningless for pairwise marginals..."** Agreed. We included the pairwise marginals just for completeness.

**"Ising model...expect that the estimation quality will degrade with the (average) interaction strength."** Thank you for this suggestion! We tested our approach in settings where the interaction strength is higher by sampling pairwise potentials from $\mathcal{N}(0,3)$ (unary potentials are still sampled from $\mathcal{N}(0,1)$). For a 10x10 grid we obtain correlations of 0.460, 0.641, and 0.770 for Mean Field, Loopy BP, and the inference network, respectively. For a 15x15 grid we analogously obtain 0.435, 0.665, and 0.738. When sampling from $\mathcal{N}(0,5)$ we obtain 0.358, 0.525, 0.619 for these 3 approaches in the 10x10 case, and 0.362, 0.459, 0.564 in the 15x15 case.

**"The experiments have in my view a preliminary character"** We agree our experiments are on small datasets. Our contribution is mostly methodological and a first step toward learning undirected models in this way, in line with early work for learning directed models with VAEs. We also note that we consider 3 fairly different archetypal settings.

**"some details of them are missing"** We apologize for the missing details, which had to be relegated to the supplementary materials due to space. We will make sure to include them should the paper be accepted.

**R2: "...having some higher order factors"...** Having too many higher order factors could be slow (since marginals scale exponentially in the order) but having a few may improve performance; we will investigate this!

**"...optimizing in the subspace..."** We explored this approach in preliminary experiments, and while it can succeed it is much less scalable: it requires (pre)computing the SVD of a large matrix (i.e., $O(|\mathcal{F}|^2)$) and we need an SVD for each graph topology (c.f., HMMs, which have different lengths per sentence). We will add more discussion around this point.

**"...hardware..."** We ran our experiments on 1080 Tis, using pytorch 1.1.

**"Why are the inference networks different..."** Empirically we observed small differences between architectures for different tasks in preliminary experiments, and we chose the best; see answer to R3 as well.

**R3: "The marginal constraints can also be enforced by matrix operations.."** See response to R2.

**"The numbers in Table 1 and Figure 2 seem inconsistent..."** The figures in Table 1 are for both unary and pairwise marginals. While LBP does slightly better for unary marginals, the inference network does much better for pairwise, and when averaged across both in Table 1, the inference network does better overall; we will clarify this further.

**"...I wonder whether some architecture of inference networks are better than others...."** We experimented with both Transformers and RNNs, and picked the architecture with the best performance in preliminary experiments. For HMMs, we find Transformers give an NLL of $\approx 116$, slightly worse than RNNs; we will include these results.

**"The form of discrepancy used ... is not clear."** We apologize for the lack of clarity: we used L2 for the synthetic experiments and Jensen-Shannon divergence for the rest, though L2 was only slightly worse. We will add this detail.

**"...random seed as a hyperparameter..."** In our experience discrete latent variable models can be quite sensitive to initialization, and so to fairly compare all model variants we randomly sample the same number of seeds for each; we do not regard this as a hyperparameter but as giving each model/method equal opportunity in experiments.

**"...did not get numbers similar..."** The following reproduces our numbers on a 1080 Ti, using pytorch 1.1, by epoch 9: `python pen_uhmm.py -cuda -K 30 -bsz 32 -dropout 0.3 -ilr 0.0003 -infarch rnnnode -init 0.001 -just_diff -lemb_size 64 -loss alt3 -lr 0.0001 -markov_order 3 -max_len 30 -not_inf_residual -optalg adam -penfunc js -q_layers 1 -qemb_size 150 -qinit 0.001 -seed 21442 -vemb_size 64 -epochs 10`. The code prints out *perplexity* rather than the NLL reported in the paper, and thus looks higher. We apologize for the potential confusion. (To convert, take $\log(\text{perplexity}) \times 50509/2747$).

**"...the authors probably ran over 100k seeds..."** We did not run 100k seeds; all methods were given 100 random configurations, as the code (line 609 in pen_uhmm.py) makes clear. Regarding stability, we calculated the std. deviation of best NLL obtained per run for each method in Table 4, giving (from top to bottom): 5.3, 0.7, 1.1, 0.5, 6.4, 2.8, 1.1. Thus amortized BFE is more stable than exact inference and LBP, but less than Mean-Field+BL and 1st Order.

[Meta-Review · NeurIPS 2019]

All reviewers agree that this is an interesting contribution to the NeurIPS community.